# Duration-of-Stay Storage Assignment under Uncertainty

**Michael Lingzhi Li**
Operation Research Center
Massachusetts Institute of Technology
Cambridge, MA 02139
mlli@mit.edu

**Elliott Wolf**
Lineage Logistics
San Francisco, California
ewolf@lineagelogistics.com

**Daniel Wintz**
Lineage Logistics
San Francisco, California
dwintz@lineagelogistics.com

## Abstract

Storage assignment, the act of choosing what goods are placed in what locations in a warehouse, is a central problem of supply chain logistics. Past literature has shown that the optimal method to assign pallets is to arrange them in increasing duration of stay (DoS) in the warehouse (the DoS method), but the methodology requires perfect prior knowledge of DoS for each pallet, which is unknown and uncertain under realistic conditions. Attempts to predict DoS have largely been unfruitful due to the multi-valuedness nature (every shipment contains identical pallets with different DoS) and data sparsity induced by lack of matching historical conditions. In this paper, we introduce a new framework for storage assignment that provides a solution to the DoS prediction problem through a distributional reformulation and a novel neural network, ParallelNet. Through collaboration with a world-leading cold storage company, we show that the system is able to predict DoS with a MAPE of 29%, a decrease of ~30% compared to a CNN-LSTM model, and suffers less performance decay into the future. The framework is then integrated into a first-of-its-kind Storage Assignment system, which is being deployed in warehouses across United States, with initial results showing up to 21% in labor savings. We also release the first publicly available set of warehousing records to facilitate research into this central problem.

## 1 Introduction

The rise of the modern era has been accompanied by ever-shortening product life cycles, straining the entire supply chain and demanding efficiency at every node. One integral part of any supply chain is warehousing (storage); warehouse operations often have major impacts downstream on the capability to deliver product on time.

One of the largest cold storage companies in the world is looking to improve the efficiency of their warehouses by optimizing the scheduling of storage systems. According to Hausman et al. (1976), the scheduling of labor in warehouses can be divided into three main components:

- *Pallet Assignment*: The assignment of multiple items to the same pallet.
- *Storage Assignment*: The assignment of pallets to a storage location.
- *Interleaving*: The overarching rules for dealing with concurrent inbound and outbound requests.

For this particular paper, we focus on the problem of storage assignment. Various papers such as Goetschalckx & Ratliff (1990) show labor efficiency to be a bottleneck. In a modern warehouse, the process of storage assignment usually involves forklift drivers moving inbound pallets from the

staging area of the warehouse to the storage location, so a sub-optimal assignment system causes unnecessary long travel times to store the pallet. Unfortunately, the inefficiency is quadrupled when the return of the forklift and the retrieval of the pallet are considered.

To increase the efficiency of the warehouse, we would thus like to minimize the total travel time needed to store a set of shipments from the staging area. Many different theoretical frameworks exist, and the details of such frameworks are contained in Appendix 9.1. The ones of chief interest are turnover-based, class-based, and Duration-of-Stay (DoS) based strategies.

Turnover-based strategies (e.g. Hausman et al. (1976), Yu & De Koster (2009)) assign locations so that the travel distance is inversely proportional to the turnover of the product. Class-based strategies (e.g. Hausman et al. (1976), Schwarz et al. (1978)) separate products into $k$ classes, with each class assigned a dedicated area of storage. DoS-based strategies (e.g. Goetschalckx & Ratliff (1990), Chen et al. (2016)) assign pallets to locations with travel distance proportional to the duration-of-stay.

Simulation experiments in Goetschalckx & Ratliff (1990) and Kulturel et al. (1999) demonstrated that under complex stochastic environments, DoS-based strategies outperform other methodologies significantly. However, among all three categories, the most commonly used strategy is class-based, as pointed out by Yu et al. (2015) and Yu & De Koster (2013). The authors and industry evidence suggest that this is due to the fact that class-based systems are relatively easy to implement, but also because DoS is not known in advance. To utilize a DoS system realistically would therefore require an accurate prediction model using the features available at shipment entry to the warehouse.

However, even with the availability of modern high-powered predictive methods including Gradient Boosted Trees and Neural Networks, there has been no documented progress in employing DoS-based methods. This reflects the following significant challenges in a dynamic, real warehouse:

- **Multi-valuedness**: It is common for $10+$ identical pallets of the same product to arrive in a single shipment, and then leave the warehouse at different times depending on the consumption of the end consumer. This causes the ground truth for the DoS of a product entering at a given time to be ill-defined.

- **Data Sparsity**: A large warehouse would have significant available historical DoS data, but such data is scattered across thousands of products/SKUs, and different operating conditions (e.g. time of the year, day of the week, shipment size). Given strong variation of DoS in a warehouse, it is very unlikely that all environment-product combinations would exist in data for the historical average to be valid for future DoS. Furthermore, new SKUs are created relatively frequently, and the predictive algorithm needs to be robust against that as well.

To solve such difficulties, we reformulate the DoS as a distribution and develop a new framework based on nonparametric estimation. Then we combine it with a parallel architecture of Residual Deep Convolutional Networks (He et al. (2015)) and Gated Recurrent Unit (GRU) networks (Cho et al. (2014)) to provide strong estimation results. As far as the authors know, this is the first documented attempt to predict DoS in warehousing systems. We further release the first public dataset of warehousing records to enable future research into this problem.

This neural network is then integrated into the larger framework, which is being implemented in live warehouses. We illustrate how initial results from the ground show appreciable labor savings.

Specifically, our contributions in this paper include:

- We develop an novel end-to-end framework for optimizing warehouse storage assignment using the distribution of DoS.
- We release a curated version of a large historical dataset of warehousing records that can be used to build and test models that predicts DoS.
- We introduce a type of neural network architecture, ParallelNet, that achieves state-of-the-art performance in estimating the DoS distribution.
- Most importantly, we present real-life results of implementing the framework with Parallel-Net in live warehouses, and show labor savings by up to $21\%$.

The structure of the paper is as followed. In Section 2, we introduce the storage assignment problem formally, and how this leads to estimating DoS in a distributional way. In Section 3, we develop the

storage assignment framework. We would introduce the dataset in Section 4 and Section 5 contains the implementation with ParallelNet, and its results compared to strong baselines. Section 6 shows the computational results, while real-life evidence is provided in Section 7.

## 2 SOLVING THE STORAGE ASSIGNMENT PROBLEM

The general storage assignment problem asks for an assignment function that outputs a storage location given a pallet and warehouse state so the total travel time for storage is minimized. We formalize such statement below and show how it naturally leads to the problem of estimating DoS.

Let warehouse $\mathcal{W}$ have locations labeled $\{1, \cdots, N\}$ where the expected travel time from the loading dock to location $i$ is $t_i$. We assume pallets arrive in discrete time periods $\{1, 2, \cdots, T\}$ where $T$ is the lifetime of the warehouse and each of the pallets has an integer DoS in $\{1, \cdots, P\}$. The warehouse uses an *assignment function* $\mathcal{A}$ that assigns each arriving pallet to an available location in $\{1, \cdots, N\}$. Define $n_i^{\mathcal{A}}(t)$ as the number of pallets (0 or 1) stored in location $i$ in time period $t$ under assignment function $\mathcal{A}$. Then our optimization problem can be stated as:

$$\min_{\mathcal{A}} \sum_{t=1}^{T} \sum_{i=1}^{N} 4 t_i n_i^{\mathcal{A}}(t) \tag{1}$$

The constant 4 is to allow for the 4 distances traveled during storage/retrieval. Given that future pallet arrivals could be arbitrary, we need additional assumptions to make the theoretical analysis tractable. In particular, we assume the warehouse is in steady state or *perfect balance*:

**Definition 1.** *Let $n_p(t)$ be the number of pallets arriving at time period $t$ that has a DoS of $p$. A warehouse is in perfect balance iff for all $p$ and $t > p$ we have $n_p(t - p) = n_p(t)$.*

This means that in each time period $t > p$, the $n_p(t - p)$ outgoing pallets with DoS of $p$ are replaced exactly by incoming pallets with DoS of $p$, so the number of pallets with DoS of $p$ in the warehouse remains constant at $z_p = \sum_{i=1}^{p} n_p(i)$ for every period $t > p$. For a real-life large warehouse, this steady state assumption is usually a good approximation except when shock events happen. Under such assumption, ideally we only need $z_p$ positions to store all the pallets with DoS of $p$, or in total $\sum_{p=1}^{P} z_p$ locations. Goetschalckx & Ratliff (1990) showed that the DoS strategy below is optimal:

**Theorem 1** (Goetschalckx & Ratliff (1990)). *Assume the warehouse is in perfect balance, and $P$ is large enough so that $z_p \leq 1$ for every $p$. Define $W(t) = \frac{\sum_{p \leq t} z_p}{\sum_{p=1}^{P} z_p}$ as the cumulative distribution of pallet DoS. Let $r = \frac{\sum_{p=1}^{P} z_p}{N}$ be the average occupancy rate of the warehouse. Then the optimal solution to (1), $\mathcal{A}^*$, would assign a pallet with DoS of $p$ to the $NrW(p)$th location.*

In other words, the locations are arranged in increasing order of DoS. We can always make $P$ large enough so that $z_p \leq 1$ by separating time periods to finer intervals. With Theorem 1, we can implement $\mathcal{A}^*$ in a real warehouse using three approximations:

1. We approximate $W(t)$ with the historical pallet cumulative DoS distribution $\hat{W}(t)$.
2. We approximate $r$ with the historical average occupancy rate of the warehouse $\hat{r}$.
3. We approximate the DoS $p$ with a prediction $\hat{p}$.

Both $\hat{W}(t)$ and $\hat{r}$ are easily retrieved from the historical data, and $N$ is known. We stress however that $\hat{W}(t)$ needs to be size-debiased. This is because pallets of DoS $p$ would appear with a frequency $\propto p z_p$ in a period of time, and thus the historical data needs to be corrected to match the definition above, which is $\propto z_p$. This is done through standard theory of size-biased distributions (Gove (2003)).

Therefore, in the next subsection, we would focus on the task of generating a prediction $\hat{p}$ of the DoS.

### 2.1 PREDICTING DURATION OF STAY

To utilize Theorem 1 above, we need to predict the DoS of a pallet at its arrival. However, in most real-life warehouses, a product $P_i$ enters the warehouse with multiple $z_i > 1$ identical pallets,

which leave at different times $t_{i1}, \cdots t_{iz_i}$. Thus, assuming the product came in at time 0, the DoS of incoming pallets of $P_i$ could be any of the numbers $t_{i1}, \cdots t_{iz_i}$, which makes the quantity ill-defined. The uncertainty can further be very large - our collaborating company had an average variance of DoS within a shipment of 10 days when the median DoS was just over 12 days.

Therefore, to account for such uncertainty, we would assume that for every shipment $S$ which contains multiple pallets, the DoS of a random pallet follows a cumulative distribution $F_S(t)$. Furthermore, we assume that such distribution can be identified using the characteristics $X_S$ of the shipment known at arrival. As in, there exists a function $g$ such that:

$$F_S(t) = g(X_S)(t)$$

for all possible shipments $S$. This assumption is not as strong as it may seem - the most important variables that affect DoS usually are the time and the good that is being transported, both of which are known at arrival. As a simple example, if the product is ice cream and it is the summer, then we expect the DoS distribution to be right skewed, as ice cream is in high demand during the summer. Moreover, the experimental results in Section 6 are indicative that $g$ exists and is highly estimable.

If the above assumption holds true, then we can estimate $g$ using machine learning. Assume we have an optimal estimate $\tilde{g}$ of $g$ relative to a loss function $l(F_S, \tilde{g}(X_S))$ denoting the loss between the actual and predicted distribution. Since by our assumption $F_S$ is identified by $X_S$, we cannot obtain any further information about DoS relative to this loss function. Thus, for each shipment with features $X_S$, we take $\hat{p}$ to be a random sample from the distribution $\tilde{F}_S = \tilde{g}(X_S)$. In the next section, we would outline our storage assignment framework based on using such prediction $\hat{p}$.

## 3 OVERVIEW OF STORAGE ASSIGNMENT FRAMEWORK

With $\hat{W}(t)$, $\hat{r}$, and $\hat{p}$ defined, we can now utilize the DoS strategy $\mathcal{A}^*$. For a pallet with predicted DoS $\hat{p}$, our storage assignment function $A : \mathbb{R} \rightarrow \mathcal{W}$ is:

$$A(\hat{p}) = \arg \min_{w \in \tilde{\mathcal{W}}} d(N\hat{r}\hat{W}(\hat{p}), w) + c(w)$$

Where $d(v, w)$ is the distance between location $v$ and $w$ in the warehouse, and $c(w)$ are other costs associated with storing at this position, including anti-FIFO orders, item mixing, height mismatch, and others. $\tilde{\mathcal{W}}$ is the set of positions that are available when the pallet enters the warehouse.

The approximate optimal position of DoS optimal location is $N\hat{r}\hat{W}(\hat{p})$. However, it is probable that such location is not ideal, either because it is not available or other realistic concerns. For the collaborating company, one important factor is item mixing due to potential cross-contamination of food allergens in close pallets. These terms are highly dependent on the specific storage company, so we include them as a general cost $c(w)$ to add to the cost $d(N\hat{r}\hat{W}(\hat{p}), w)$ of not storing the pallet at the DoS optimal position. The resulting location is chosen based on the combination of the two costs.

In summary, our framework consists of three steps:

1. Utilize machine learning to provide an estimate $\tilde{g}$ of $g$ with respect to some loss function $l$.

2. For a shipment $S$, calculate $\tilde{F}_S = \tilde{g}(X_S)$ and generate a random sample $\hat{p}$ from $\tilde{F}_S$.

3. Apply the assignment function $A$ defined above to determine a storage location $A(\hat{p})$.

## 4 WAREHOUSING DATASET OVERVIEW AND CONSTRUCTION

In this section, we introduce the historical dataset from the cold storage company to test out the framework and model introduced in Section 3.

### 4.1 OVERVIEW OF THE DATA

The data consists of all warehouse storage records from 2016.1 to 2018.1, with a total of 8,443,930 records from 37 different facilities. Each record represents a single pallet and one shipment of goods usually contain multiple identical pallets (which have different DoS). On average there are 10.6 pallets per shipment in the dataset. The following covariates are present:

- **Non-sequential Information**: Date of Arrival, Warehouse Location, Customer Type, Product Group, Pallet Weight, Inbound Location, Outbound Location

- **Sequential Information**: Textual description of product in pallets.

Inbound and Outbound location refers to where the shipment was coming from and will go to. The records are mainly food products, with the most common categories being (in decreasing order): chicken, beef, pork, potato, and dairy. However, non-food items such as cigarettes are also present.

The item descriptions describe the contents of the pallet, but most of them are not written in a human readable form, such as "NY TX TST CLUB PACK". Acronyms are used liberally due to the length restriction of item descriptions in the computer system. Furthermore, the acronyms do not necessarily represent the same words: "CKN WG" means "chicken wing" while "WG CKN" means "WG brand chicken". Therefore, even though the descriptions are short, the order of the words is important.

To enable efficient use of the descriptions, we decoded common acronyms by hand (e.g. tst → toast). However, the resulting dataset is not perfectly clean (intentionally so to mimic realistic conditions) and contains many broken phrases, misspelled words, unidentified acronyms, and other symbols.

## 4.2 PUBLIC RELEASE VERSION [1]

We release the above dataset, which as far as the authors know, is the first publicly available dataset of warehousing records.

The collaborating company transports an appreciable amount ($> 30\%$) of the entire US refrigerated food supply, so US law prohibits the release of the full detail of transported shipments. Furthermore, NDA agreements ban any mentioning of the brand names. Thus, for the public version, we removed all brands and detailed identifying information in the item descriptions. The testing in the section below is done on the private version to reflect the full reality in the warehouse, but the results on the public version are similar (in Appendix 9.4) and the conclusions carry over to the public version.

## 5 IMPLEMENTING THE FRAMEWORK

The Framework described in Section 3 requires the knowledge of four parameters: $X_S$ and $F_S$ for pallets in the training data, the loss function $l(F_S, \tilde{F}_S)$, and the machine learning model estimate $\tilde{g}$.

$X_S$ is immediately available in the form of the non-sequential and sequential information. For the textual description, we encode the words using GloVe embeddings. (Pennington et al. (2014)) We limit the description to the first five words with zero padding.

For $F_S$, we exploit that each shipment arriving usually contains $p \gg 1$ units of the good. We treat these $p$ units as $p$ copies of a one-unit shipment, denoted $S_1, \cdots S_p$. Then by using the DoS for each of these $p$ units $(T_1, \cdots T_p)$, we can create an empirical distribution function $\hat{F}_S(t) = \frac{1}{p} \sum_{i=1}^{n} \mathbf{1}_{T_i \leq t}$ for the DoS of the shipment. This is treated as the ground truth for the training data.

To obtain a loss function for the distribution, we selected the $5\%, \cdots 95\%$ percentile points of the CDF, which forms a 19-dimensional output. This definition provides a more expressive range of CDFs than estimating the coefficients for a parametric distribution, as many products' CDF do not follow any obvious parametric distribution. Then we chose the mean squared logarithmic error (MSLE) as our loss function to compare each percentile point with those predicted. This error function is chosen as the error in estimating DoS affects the positioning of the unit roughly logarithmically in the warehouse under the DoS policy. For example, estimating a shipment to stay 10 days rather than the true value of 1 day makes about the same difference in storage position compared to estimating a shipment to stay 100 days rather than a truth of 10. This is due to a pseudo-lognormal distribution of the DoS in the entire warehouse as seen in the historical data.

---

[1] Academic users can currently obtain the dataset by inquiring at dwintz@lineagelogistics.com. It would be hosted online in the near future.

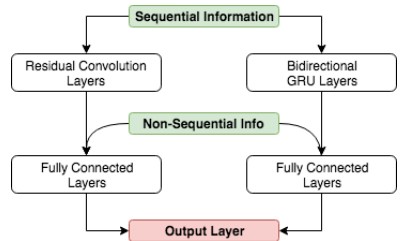

Figure 1: ParallelNet Simplified Architecture. Green boxes are inputs and the red box is the output. We separate inputs into sequential data and non-sequential data to exploit different types of data.

Thus, our empirical loss function is defined as:

$$L(\hat{F}_S, \tilde{F}_S) = \frac{1}{19} \sum_{i=1}^{19} \left( \log(\hat{F}_S^{-1}(0.05i) + 1) - \log(\tilde{F}_S^{-1}(0.05i) + 1) \right)^2$$

Now, we would introduce the machine learning algorithm $\tilde{g}$ to approximate $F_S$.

## 5.1 INTRODUCTION OF PARALLELNET

For the dataset introduced in Section 4, the textual description carries important information relating to the DoS distribution. The product identity usually determines its seasonality and largely its average DoS, and therefore it would be desirable to extract as much information as possible through text.

We would utilize both convolutional neural networks (CNN) and recurrent neural networks (RNN) to model the textual description. As illustrated in Section 3, word order is important in this context, and RNNs are well equipped to capture such ordering. As the textual information is critical to the DoS prediction, we would supplement the RNN prediction with a CNN architecture in a parallel manner, as presented in Figure 1. We then utilized a residual output layer to produce 19 percentile points $(\tilde{F}_S^{-1}(0.05), \tilde{F}_S^{-1}(0.1), \cdots, \tilde{F}_S^{-1}(0.95))$ that respect the increasing order of these points.

This architecture is similar to ensembling, which is well known for reducing bias in predictions (see Opitz & Maclin (1999)). However, it has the additional advantage of a final co-training output layer that allows a complex combination of the output of two models, compared to the averaging done for ensembling. This is particularly useful for our purpose of predicting 19 quantile points of a distribution, as it is likely the CNN and RNN would be better at predicting different points, and thus a simple average would not fully exploit the information contained in both of the networks. We would see in Section 6 that this allows the model to further improve its ability to predict the DoS distribution. We also further note that this is similar to the LSTM-CNN framework proposed by Donahue et al. (2014), except that the LSTM-CNN architecture stakcs the CNN and RNN in a sequential manner. We would compare with such framework in our evaluation in Section 6.

In interest of brevity, we omit the detailed architecture choice in RNN and CNN along with the output layer structure and include it in Appendix 9.2. Hyperparameters are contained in Appendix 9.3.

## 6 COMPUTATION RESULTS

In this section, we test the capability of ParallelNet to estimate the DoS distributions $F_S$ on the dataset introduced in Section 4. We separate the dataset introduced in Section 4 into the following:

- Training Set: All shipments that exited the warehouse before 2017/06/30, consisting about 60% of the entire dataset.

- Testing Set: All shipments that arrived at the warehouse after 2017/06/30 and left the warehouse before 2017/07/30, consisting about 7% of the entire dataset.

- Extended Testing Set: All shipments that arrived at the warehouse after 2017/09/30 and left the warehouse before 2017/12/31, consisting about 14% of the entire dataset.

We then trained five separate neural networks and two baselines to evaluate the effectiveness of ParallelNet. Specifically, for neural networks, we evaluated the parallel combination of CNN and RNNs against a vertical combination (introduced in Donahue et al. (2014)), a pure ensembling model, and the individual network components. We also compare against two classical baselines, gradient-boosted trees (GBM) (Friedman (2001)) and linear regression (LR), as below:

- **CNN-LSTM** This implements the model in Donahue et al. (2014). To ensure the best comparability, we use a ResNet CNN and a 2-layer GRU, same as ParallelNet.
- **CNN+LSTM** This implements an ensembling model of the two architectures used in ParallelNet, where the two network's final output is averaged.
- **ResNet (CNN)** We implement the ResNet arm of ParallelNet.
- **GRU** We implement the GRU arm of ParallelNet.
- **Fully-connected Network (FNN)** We implement a 3-layer fully-connected network.
- **Gradient Boosted Trees with Text Features (GBM)** We utilize Gradient Boosted Trees (Friedman, 2001) as a benchmark outside of neural networks. As gradient boosted trees cannot self-generate features from the text data, we utilize 1-gram and 2-gram features.
- **Linear Regression with Text Features (LR)** We implement multiple linear regression. Similar to GBM, we generate 1-gram and 2-gram features. The $y$ variable is log-transformed as we assume the underlying duration distribution is approximately log-normal.

All neural networks are trained on Tensorflow 1.9.0 with Adam optimizer (Kingma & Ba, 2014). The learning rate, decay, and number of training epochs are 10-fold cross-validated. The GBM is trained on R 3.4.4 with the lightgbm package, and number of trees 10-fold cross-validated over the training set. The Linear Regression is trained with the lm function on R 3.4.4. We used a 6-core i7-5820K, GTX1080 GPU, and 16GB RAM. The results on the Testing Set are as followed:

| Architecture | Testing Set | | Extended Testing Set | |
|---|---|---|---|---|
| | MSLE | MAPE | MSLE | MAPE |
| ParallelNet | 0.4419 | 29% | 0.7945 | 51% |
| CNN-LSTM | 0.4812 | 41% | 0.9021 | 80% |
| CNN+LSTM | 0.5024 | 47% | 0.9581 | 91% |
| CNN | 0.6123 | 70% | 1.0213 | 124% |
| GRU | 0.5305 | 47% | 1.1104 | 122% |
| FNN | 0.8531 | 120% | 1.0786 | 130% |
| GBM | 1.1325 | 169% | 1.2490 | 187% |
| LR | 0.9520 | 132% | 1.1357 | 140% |

Table 1: Table of Prediction Results for Different Machine Learning Architectures

| Architecture | Testing Set | |
|---|---|---|
| | MSLE | MAPE |
| ParallelNet | 0.4419 | 29% |
| Without Product Group | 0.6149 | 65% |
| Without Date of Arrival | 0.5723 | 61% |
| Without Customer Type | 0.5687 | 58% |
| Without All Nontextual Features | 0.8312 | 110% |
| Without Textual Features | 0.9213 | 125% |

Table 2: Ablation Studies

We can see that ParallelNet comfortably outperforms other architectures. Its loss is lower than the vertical stacking CNN-LSTM, by $8\%$. The result of $0.4419$ shows that on average, our prediction in the 19 percentiles is $44\%$ away from the true value. We also note that its loss is about $15\%$ less than the pure ensembling architecture, indicating that there is a large gain from the final co-training layer.

We also note that the baselines (GBM, LR) significantly underperformed neural network architectures, indicating that advanced modeling of text is important. We also conducted ablation studies to find that product group, date of arrival, and customer type are important for the task, which is intuitive. Furthermore, both nontextual and textual features are required for good performance.

We then look at a different statistic: the Median Absolute Percentage Error (MAPE). For every percentile in every sample, the Absolute Percentage Error (APE) of the predicted number of days $\hat{T}$ and the actual number of days $T$ is defined as:

$$\text{APE}(\hat{T}, T) = \frac{|\hat{T} - T|}{T}$$

Then MAPE is defined as the median value of the APE across all 19 percentiles and all samples in the testing set. This statistic is more robust to outliers in the data.

As seen in Table 2, ParallelNet has a MAPE of $29\%$. This is highly respectable given the massive innate fluctuations of a dynamic warehouse, as this means the model could predict $50\%$ of all

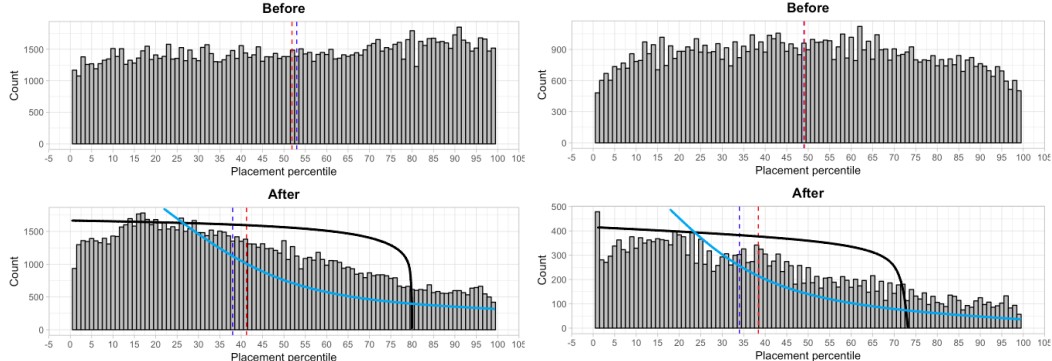

Figure 2: Placement percentile of putaway pallets before and after using a DoS system for 5 months in two selected facilities. Red line denotes the mean and blue line denotes the median.

percentiles with an error less than $29\%$. The result also compares well with the other methods, as ParallelNet reduces the MAPE by $29.3\%$ when compared to the best baseline of CNN-LSTM.

If we look further out into the future in the Extended Testing Set, the performance of all algorithms suffer. This is expected as our information in the training set is outdated. Under this scenario, we see that ParallelNet still outperforms the other comparison algorithms by a significant amount. In fact, the difference between the pairs (CNN-LSTM, ParallelNet) and (CNN+LSTM, ParallelNet) both increase under the MSLE and MAPE metrics. This provides evidence that a parallel co-training framework like that of ParallelNet is able to generalize better. We hypothesize that this is due to the reduction in bias due to ensembling-like qualities leading to more robust answers.

## 7    REAL-LIFE IMPLEMENTATION RESULTS

With the favorable computational results, the collaborating company is implementing the framework with ParallelNet across their warehouses, and in this section we would analyze the initial results.

The graphs in Figure 2 records the empirical distribution of the placement percentiles before and after the DoS framework was online. The placement percentile is the percentile of the distance from the staging area to the placement location. Thus, 40% means a pallet is put at the 40th percentile of the distance from staging. The distance distribution of locations is relatively flat, so this is a close proxy of driving distance between staging and storage locations, and thus time spent storing the pallets.

Additionally, we plotted two curves obtained by simulating placement percentiles under a perfect DoS strategy (blue) and a greedy strategy that allocates the closest available position to any pallet ignoring DoS (black). These simulations assumed all locations in the warehouse are open to storage and ignored all other constraints in the placement; thus they do not reflect the full complex workings of a warehouse. However, this simple simulation show clearly the advantages of the DoS strategy as it is able to store more pallets in the front of the warehouse. We observe that in both facilities, the real placement percentiles behaved relatively closely to the blue curve. This is strong evidence that the implemented DoS framework is effective in real-life and provides efficiency gains. We note that there is a drop in the lower percentiles compared to the DoS curve - this is due to some locations close to the staging area being reserved for packing purposes and thus not actually available for storage.

Specifically, Facility A had an average placement percentile of $51\%$ before, and $41\%$ after, while Facility B had an average placement percentile of $50\%$ before, and $39\%$ after. On average, we record a $10.5\%$ drop in absolute terms or $21\%$ in relative terms. This means that the labor time spent on storing pallets has roughly declined by $21\%$. An unpaired $t$-test on the average putaway percentile shows the change is statistically significant on the $1\%$ level for both facilities. This provides real-life evidence that the system is able to generate real labor savings in the warehouses.

## 8 CONCLUSION AND REMARKS

In conclusion, we have introduced a comprehensive framework for storage assignment under an uncertain DoS. We produced an implementation of this framework using a parallel formulation of two effective neural network architectures. We showed how the parallel formulation has favorable generalization behavior and out-of-sample testing results compared with sequential stacking and ensembling. This has allowed this framework to be now implemented in live warehouses all around the country, and results show appreciable labor savings on the ground. We also release the first dataset of warehousing records to stimulate research in this central problem for storage assignment.

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

# 9 APPENDIX

## 9.1 LITERATURE REVIEW ON STORAGE ASSIGNMENT

Since Hausman et al. (1976), many different theoretical frameworks have been introduced, which can roughly be separated into two classes: dedicated storage systems and shared storage systems.

### 9.1.1 DEDICATED STORAGE SYSTEMS

For this class of storage systems, each product gets assigned fixed locations in the warehouse. When the product comes in, it is always assigned to one of the pre-determined locations. Under this constraint, it is optimal to dedicate positions with travel distance inversely proportional to the turnover of the product, as shown in Goetschalckx & Ratliff (1990). Turnover of a product is defined as the inverse of Cube per Order Index (COI), which is the ratio of the size of the location it needs to the frequency of retrieval needed. Heuristically, those products with the smallest surface footprint and the highest frequency should be put closest to the warehouse entry, so that those locations are maximally utilized.

### 9.1.2 SHARED STORAGE SYSTEMS

This class of storage systems allows multiple pallets to occupy the same position in the warehouse (at different times). It is widely considered to be superior than dedicated storage systems due to its savings on travel time and smaller need for storage space, as shown by Yu et al. (2015), Malmborg (2000). Within this category, there are mainly three strategies:

- *Turnover (Cube-per-Order, COI) Based*: Products coming into the warehouse are assigned locations so that the resultant travel distance is inversely proportional to the turnover of the product. Examples of such work includes Hausman et al. (1976), Yu & De Koster (2009), and Yu & De Koster (2013).

- *Class Based*: Products are first separated into $k$ classes, with each class assigned a dedicated area of storage. The most popular type of class assignment is called ABC assignment, which divides products into three classes based on their turnover within the warehouse. Then within each class, a separate system is used to sort the pallets (usually random or turnover-based). It was introduced by Hausman et al. (1976), and he showed that a simple framework saves on average $20 - 25\%$ of time compared to the dedicated storage policy in simulation. Implementation and further work in this area include Rosenblatt & Eynan (1989), Schwarz et al. (1978) and Petersen et al. (2004).

- *Duration-of-Stay (DoS) Based*: Individual products are assigned locations with travel distance proportional to the duration of stay. Goetschalckx & Ratliff (1990) proved that if DoS is known in advance and the warehouse is completely balanced in input/output, then the DoS policy is theoretically optimal. The main work in this area was pioneered by Goetschalckx & Ratliff (1990). Recently, Chen et al. (2016) and Chen et al. (2010) reformulated the DoS-based assignment problem as a mixed-integer optimization problem in an automated warehouse under different configurations. Both papers assume that the DoS is known exactly ahead of time.

### 9.2 ARCHITECTURE CHOICE

In modeling text, there are three popular classes of architectures: convolutional neural networks (CNN), recurrent neural networks (RNN), and transformers. In particular recently transformers have gained popularity due to their performance in machine translation and generation tasks (e.g. Devlin et al. (2018), Radford et al.). However, we argue that transformers are not the appropriate model for the textual descriptions here. The words often do not form a coherent phrase so there is no need for attention, and there is a lack of long-range dependency due to the short length of the descriptions.

#### 9.2.1 BIDIRECTIONAL GRU LAYERS

Gated Recurrent Units, introduced by Cho et al. (2014), are a particular implementation of RNN intended to capture long-range pattern information. In the proposed system, we further integrate bi-directionality, as detailed in Schuster & Paliwal (1997), to improve feature extraction by training the sequence both from the start to end and in reverse.

The use of GRU rather than LSTM is intentional. Empirically GRU showed better convergence properties, which has also been observed by Chung et al. (2014), and better stability when combined with the convolutional neural network.

#### 9.2.2 RESNET CONVOLUTIONAL LAYERS

In a convolutional layer, many independent filters are used to find favorable combinations of features that leads to higher predictive power. Further to that, they are passed to random dropout layers introduced in Srivastava et al. (2014) to reduce over-fitting and improve generalization. Dropout layers randomly change some outputs in $c_0, \cdots c_i$ to zero to ignore the effect of the network at some nodes, reducing the effect of over-fitting.

Repeated blocks of convolution layers and random dropout layers are used to formulate a deep convolution network to increase generalization capabilities of the network.

However, a common problem with deep convolutional neural networks is the degradation of training accuracy with increasing number of layers, even though theoretically a deeper network should perform at least as well as a shallow one. To prevent such issues, we introduce skip connections in the convolution layers, introduced by Residual Networks in He et al. (2015). The residual network introduces identity connections between far-away layers. This effectively allows the neural network to map a residual mapping into the layers in between, which is empirically shown to be easier to train.

#### 9.2.3 OUTPUT LAYER

We designed the output layer as below in Figure 3 to directly predict the 19 percentile points $(\tilde{F}_S^{-1}(0.05), \tilde{F}_S^{-1}(0.1), \cdots, \tilde{F}_S^{-1}(0.95))$:

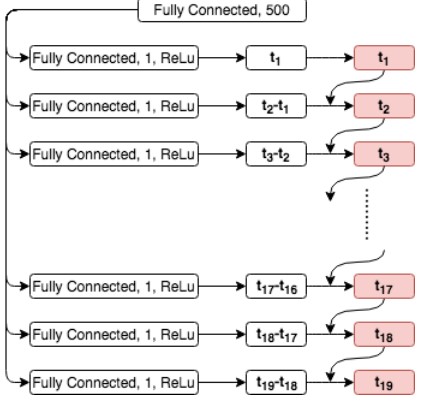

Figure 3: Output Layer. Here $t_1, \cdots t_{19}$ corresponds to the $5\%, \cdots 95\%$ percentile points.

Note that the 19 percentile points are always increasing. Thus the output is subjected to 19 separate 1-neuron dense layers with ReLu, and the output of the previous dense layer is added to the next one, creating a *residual output layer* in which each (non-negative) output from the 1-neuron dense layer is only predicting the residual increase $t_{i+1} - t_i$.

## 9.3 DETAILED SETTINGS OF IMPLEMENTED ARCHITECTURE

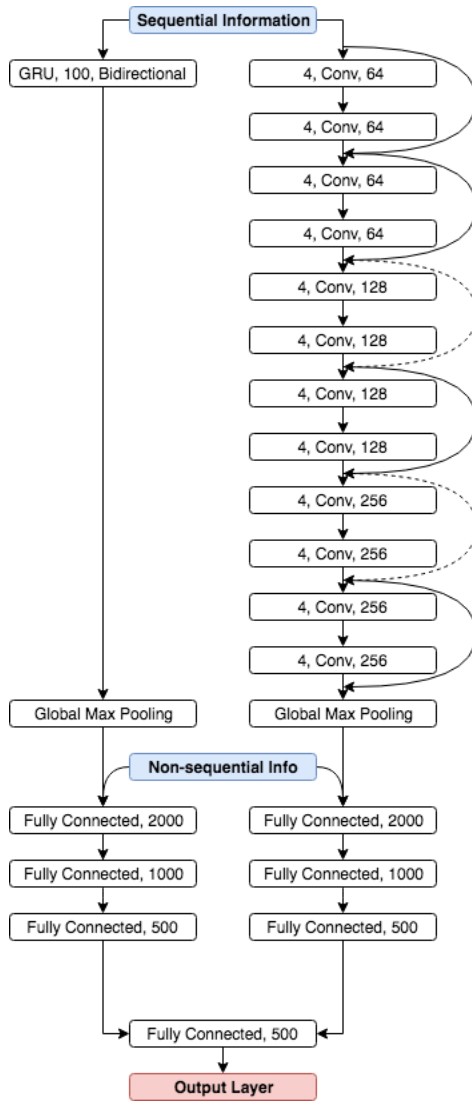

Figure 4: ParallelNet Architecture

## 9.4 TESTING RESULTS ON PUBLIC DATASET

| Architecture | Testing Set | | Extended Testing Set | |
|---|---|---|---|---|
| | MSLE | MAPE | MSLE | MAPE |
| ParallelNet | 0.4602 | 34% | 0.7980 | 50% |
| CNN | 0.6203 | 73% | 1.0087 | 122% |
| GBM | 1.1749 | 175% | 1.2605 | 190% |

Table 3: Results on Public Dataset for Selected Architectures

