# OpenReview forum: "Duration-of-Stay Storage Assignment under Uncertainty"
_ICLR.cc/2020/Conference — Accept (Spotlight)_

### Official Review · AnonReviewer1 · 2019-10-13
**Official Blind Review #1**

**Rating:** 6

**Review:**

This paper introduces a model to predict the duration of stay of goods in a warehouse, and releases an extensive dataset of warehousing records used in the experiments.

I found the paper very interesting to read, and I believe it could serve as an inspiration to researches more interested in applied ML, since is shows that relatively standard architectures can make a big impact in solving real-world problems.

I have some questions on the experiments:
- The model used in the experiments is a combination of a CNN and a GRU architecture. One thing that is not clear to me is why the author chose to limit the sequential input to the model to only 5 words (in particular in the GRU branch).
- Did you make any analysis on which information is more important for the model when making predictions? Is the sequential information describing the type of product dominating the non-sequential information?
- What are the generalization/transferability properties of the model across different locations?
- Based on the results on the extended training set it seems clear that due to the temporal shift in the data the model will have to be retrained fairly often. Did you experiment with this in your real-life implementation?

To me, the released dataset is one of the key contributions of this work. However, I have some doubts on the reproducibility of the experiments.
- Due to regulations/NDA you were not able to release some details of the data that seem really important to me. My guess is for example that the brand name is a key information to have when predicting DoS. Is the data really meaningful even without it?
- To verify the above question as well as to create a baseline for people interested in using the public dataset, you should add your results on the public version of dataset (even in appendix).

Overall I liked the paper so I argue for acceptance, provided that I receive a satisfactory answer to the above concerns.

Small typo:
- Assumption 1: ".. exists a function f" -> ".. exists a function g"


**Experience Assessment:**

I have published one or two papers in this area.

**Review Assessment: Checking Correctness Of Derivations And Theory:**

I assessed the sensibility of the derivations and theory.

**Review Assessment: Checking Correctness Of Experiments:**

I assessed the sensibility of the experiments.

**Review Assessment: Thoroughness In Paper Reading:**

I read the paper at least twice and used my best judgement in assessing the paper.

---

> ### Author Response · Authors · 2019-11-06
> **Response to Review 1**
>
> Thank you Reviewer #1 for your comments.
>
> 1. The vast majority of the labels (after cleaning) are 4-5 words, and those that are significantly longer are usually due to data issues rather than having any significant information.
>
> 2. We found both the sequential information and nonsequential information to be useful in most of the pallet types (some product groups contain information about storage types that are not contained in text) but for those that are less predictable (in terms of DoS), the non-sequential information is more informative, and for those that have more predictable (less MAPE), the sequential information is more informative.
>
> 3. Currently, we are retraining each model for each facility as each has vastly different characteristics and product type. However, we agree it would be interesting to explore work in transfer learning for this. The dataset provided would be capable of investigating this (the facility number is provided).
>
> 4. After some experimentation, we are currently retraining the model every 3 months to keep up with the current trend. The model has roughly been implemented for 6 months now and we have not seen major issues with such schedule.
>
> Regarding data:
>
> The brand name is actually (probably surprisingly) not key to prediction of DoS. This is because we have customer type + product group information that is separate from the textual data, so the removal of the brand name results in quite little loss, and usually the brand name means little in English.
>
> We will add results on the public version of the dataset in the appendix.

---

> > ### Comment · AnonReviewer1 · 2019-11-09
> > **Thanks for the clarifications**
> >
> > Thanks for the  reply, I will keep my vote unchanged and will argue for acceptance

---

> > > ### Author Response · Authors · 2019-11-10
> > > **Paper Revised**
> > >
> > > Once again, thank you Reviewer #1 for your helpful comments and support.
> > >
> > > We have significantly revised the paper in response to all the comments, and incorporated your comments into the new version. Specifically, we added testing results on the public dataset. As you would be able to see, the results are extremely similar and the loss of brand information is not material to the whole prediction.
> > >
> > > Also, thank you for pointing out the typo - we have also corrected that.
> > >
> > > Best
> > > -Authors of #924

---

### Official Review · AnonReviewer3 · 2019-10-21
**Official Blind Review #3**

**Rating:** 3

**Review:**

The paper tackles the problem of assigning pallets to storage positions in a warehouse so as to minimize total travel time. Previous literature has shown that if durations of stay (DoS) is known in advance then the optimal storage position is inversely proportional to this. The paper proposes to predict a DoS distribution for each pallet using a 19 point CDF with a neural network. The authors show that the proposed network is better than a few other neural baselines and that it can predict the DoS reasonably well.
The authors report significant positive real life results in two warehouses and releases their dataset.

The paper is interesting and seems to make significant progress on an important real-life issue. The release of a large realistic dataset is a major contribution to this field.

Despite this I will not recommend this paper for publishing at ICLR, simply because I think it falls outside the scope of the conference.

I would recommend the authors seek to publish in a venue more focused on logistics or operational research (OR), where I think the paper could have a great impact.

I would have appreciated a more in-depth description of the problem. It seems that the general problem would be to minimize the expected cost (distance moved) over the lifetime of the warehouse. Under what simplifying assumptions does that reduce to minimizing the expected cost of each pallet individually as done here?

Similarly, I would have preferred a more formal definition of the loss, e.g. start from the expected distance moved and derive the (approximate) loss from there. Approximations are fine as long as you tell the reader what you're approximating, why you need to do it, and what the biases are.

I would also move the detailed description and figures of the actual neural network to the appendix since it's not central to the understanding of the paper (aside from the CDF formulation).


------------ UPDATE ------------

The area chair didn't get back to me, but I'll assume the paper is in scope and as such have changed my rating to weak reject.

I'd prefer if the paper spend less space arguing that a specific network architecture is the right choice and prioritize introducing the problem more formally. I would move the simulations of different storage strategies (figure 2, blue and black lines) into the introduction, as part of introducing the reader to the problem and evaluation, and explain how figure 2 is related to what you really want to minimize: "total pallet distance moved over time x". I wonder if the pallet percentile distribution (figure 2) is actually interesting, other than as a way to compute the "total pallet distance moved over time x".

I fear the authors feel the need to introduce a "novel" NN in order to publish at ICLR and thus spend a lot of effort describing something that is really not very interesting (even giving it a name). Combining textual and non-textual features is not a new concept, and combining a RNN layer and a CNN layer is not novel. I'm skeptical that this exact network structure is needed. It's clear you should combine the textual and non-textual features, but is this *exact* network structure needed? I would find that very surprising. How about MLP(concat(CNN(text), RNN(text), non-text))?

Another gripe is the heuristic nature of the loss function. It's clear predicting the DoS distribution better is the right thing to do, but is the sum of L2 distances to the emprical log CDF the *right* measure or a heuristic? What is the right thing to do? What is it approximating if anything? Why?
I would expect the right thing to do is to maximize the probability of the observed duration of stay under the model, i.e max_\theta p(DoS|x, \theta), i.e. min_\theta \sum_i -log(p(DoS_i|x_i, \theta). If this is log-normally distributed, then a log-normal distribution is probably a good distribution. Or perhaps a mixture of log-normals. I think you can even use the 19 point CDF output layer and evaluate the data likelihood under this, but I *don't* think the 19 point L2 distance to the empirical probability distributions is the correct thing to do. Is that even defined if there's less than 19 pallets with the same inputs x?

**Experience Assessment:**

I do not know much about this area.

**Review Assessment: Checking Correctness Of Derivations And Theory:**

I assessed the sensibility of the derivations and theory.

**Review Assessment: Checking Correctness Of Experiments:**

I assessed the sensibility of the experiments.

**Review Assessment: Thoroughness In Paper Reading:**

I read the paper at least twice and used my best judgement in assessing the paper.

---

> ### Author Response · Authors · 2019-11-06
> **Response to Review 3**
>
> Thank you Reviewer #3 for your comments.
>
> We understand that this is not a "traditional" ML/DL paper, and might feel out of place. However, I would like to quote directly from the ICLR website:
>
> "We consider a broad range of subject areas including feature learning, metric learning, compositional modeling, structured prediction, reinforcement learning, and issues regarding large-scale learning and non-convex optimization, as well as applications in vision, speech recognition, text understanding, games, music, computational biology, and others."
>
> The applications of ML are considered under the scope of ICLR, and there has been many past application-focused papers in music, biology, audio, and others. Accepted papers (in which application is the main contribution) include, from 2019 alone:
>
> https://openreview.net/forum?id=HJGkisCcKm
> https://openreview.net/forum?id=ByMVTsR5KQ
> https://openreview.net/forum?id=SkloDjAqYm
>
> We feel like drawing a line at logistics feels arbitrary.
>
> Traditional industries such as warehousing have been under-represented at this conference, but we believe that such situation makes our paper more and not less important. We aim to provide a solid demonstration of the potential opportunity deep learning has in logistics and to stimulate more research into this field that has been largely neglected as an application area in deep learning. We further provide the first-of-its-kind dataset so that research into this field can be jump started.
>
> The reasons for choosing this conference and not a logistics/OR conference are:
>
> 1. This is a real-life demonstration of the power of deep learning.
>
> 2. We hope this paper can stimulate research into this field from deep learning to develop techniques that are suitable to this field (e.g. as we argued in this paper, reinforcement learning cannot be immediately applied).
>
> 3. ICLR permits us to reach a wider range of audience, and help turn around the nature that applications in these traditional industries are often neglected.
>
> We sincerely hope you would reconsider your position on this.

---

> > ### Comment · AnonReviewer3 · 2019-11-06
> > **Response**
> >
> > Thank you for your thoughtful response. I've asked the Area chair whether she/he thinks this paper is within the scope of ICLR. I'll get back to you later once I get a reply.

---

> > > ### Author Response · Authors · 2019-11-08
> > > **Response**
> > >
> > > Thank you Reviewer #3 for reaching out to the Area Chair. Please do update us when you have a reply.
> > >
> > > Here is an initial response to your other comments:
> > >
> > > 1. You are correct that the general objective would be to minimize the expected distance moved over the lifetime of the warehouse. Goetschalckx & Ratliff (1990) details the theoretical argument in how to show that DoS strategy (organize pallets by Duration of Stay) falls out as the optimal strategy under certain simplifying conditions.
> > >
> > > Specifically, one of the main assumptions is that the warehouse is in perfect balance (one shipment of good i leaves just as one shipment of good i arrives). Of course such assumption is not perfectly satisfied in real-life, but in a highly dynamic warehouse the time between a shipment of good i leaving and a shipment of good i arriving is small so we believe this is a good approximation. We are happy to add additional descriptions about the conditions under which DoS is proved to be optimal into the paper, but would leave out the specific proof and refer to the reference above.
> > >
> > > Then, the idea of the cost function is to measure how "far" away we are from the theoretical optimal solution of DoS, by using the squared error loss.
> > >
> > > 2. Thank you for the suggestion of moving the architecture into the appendix. We completely agree with your main point, and that was our main motivation to move the description of the CNN and RNN into the appendix. The reason for leaving a partial description in the main paper is to illustrate the "smart ensembling" between CNN and RNN. We feel although this is a combination of two existing architectures, the "smart ensembling" is relatively novel, and more importantly the fact that it significantly outperforms simple ensembling in the experiments could be of use to practitioners in the future.

---

> > > > ### Comment · AnonReviewer3 · 2019-11-13
> > > > **Update**
> > > >
> > > > Thank you for the thoughtful response.
> > > >
> > > > The Area chair didn't get back to me, but I've updated my review assuming this is in scope for ICLR.

---

> ### Comment · AnonReviewer1 · 2019-11-09
> **This paper could be interesting for the ICLR community**
>
> Hi, when first reviewing the paper I had similar concerns on whether ICLR is a good fit for this paper. In the end I think it is:
> - as the authors replied to your review, the ICLR call for papers clearly states that application papers are welcome.
> - even more importantly to me however, as I wrote in my review this paper "could serve as an inspiration to researches more interested in applied ML, since is shows that relatively standard architectures can make a big impact in solving real-world problems."

---

> ### Author Response · Authors · 2019-11-10
> **Paper Revised**
>
> Once again, thank you Reviewer #3 for the helpful comments.
>
> We have significantly updated the paper and incorporated your comments. This includes:
>
> 1. We added the conditions under which Goetschalckx and Ratliff (1990) proved DoS strategy is optimal. We have opted not to include a full derivation of their proof as we think it would deviate from the main message of the paper.
>
> 2. In response to your comment, we moved the discussion on the output layer into the appendix. We have left the discussion about the "smart ensembling" structure in the main text so that future practitioners can find it useful.
>
>
> We sincerely hope you can update your review in light of these changes (and also the Area Chair's feedback on suitability).
>
> Best
> -Authors of #924

---

> ### Author Response · Authors · 2019-11-13
> **Response to Reviewer #3**
>
> We sincerely thank you Reviewer 3 for updating the review. Here is our response to your new comments:
>
> 1. We would like to kindly point Reviewer #3 toward Appendix #1 which was included a detailed rundown of the different storage strategies one can use, and painstakingly studied by the storage assignment literature. The reason for directly arguing that DoS prediction is the right way to go is to avoid detracting the paper from its main point which is "DoS strategy helps storage assignment a lot, and using deep learning can (finally) make it happen". We are happy to add a graph about this into the Appendix.
>
> 2. We want to categorically and completely deny that this network was designed to "publish at ICLR". Note that this is not a task in which one does testing on some data and report results. This network needs to be implemented to affect more than 30% of the US refrigerated food supply (and, in fact, is implemented) and if an off-the-shelf algorithm gave the best results, we would have surely utilized it. This is not a question of whether we are trying to gain "novelty".
>
> 3. We do not claim this is the "exact" network needed. We are more than happy for future research to show a better performing architecture - that's the point of research :) Note that however we have done ablation experiments to show that this network outperforms even a very closely related network (CNN-LSTM) which has the same amount of computational complexity as the current network. To give it a name is to provide a reference to the neural network at hand and to prevent confusion with similar networks (aka originally we wanted to call it CNN-LSTM, but that was already taken by the comparison algorithm in the paper). We actually also tested the structure you proposed - it performs better than the comparison CNN-LSTM but slightly less than the current one. The reason for not including it in the experiments is to avoid confusion by introducing more variations of similar structure. We also want to again stress that since this needs to be applied to real warehouses, at some point we had to stop exploring and determined which one we would use (it is not possible to update architectures fast in warehouses).
>
> 4. We agree with Reviewer #3 that our network is not very novel - we do not claim so otherwise. As we stress, this is mainly an application paper and we would very much appreciate if the reviewer would calibrate the review metrics based on other application paper accepted at ICLR. We spent in total 0.75 pages discussing about the network so we do not believe it is a particularly long discussion. The main point of that 0.75 pages is to point out certain ensembling techniques that we utilized which turned out to be extremely helpful in the real-life experiments. Note in the experiments we showed that a simple ensembling of the two networks underperformed ours very significantly. This might not be theoretically very interesting but application wise useful to people who could explore different directions.
>
> 5. Thank you for pointing out using MLE to optimize the distribution. We have actually attempted to do so and did not succeed, so we wrote in the paper that our current nonparametric point estimation "provides a more expressive range of CDFs than estimating the coefficients for a parametric distribution." The reason is that the lognormal distribution is only approximately true when we combine all shipments of different products together. Individually, they fall far from a lognormal distribution or even a mixture of lognormal distributions, or really, any classical parametric function family that we tried. We will update the paper and be more clear about this.
>
> For the vast majority (>99%) of shipments, there are many more than 19 pallets of them (one shipment contains a lot of pallets for each product). For those that have less than 19 pallets, we still estimate their empirical CDF (which is defined) using our formulation but concur that it may not be as accurate. This has not proven to be a problem in our actual implementation.

---

> > ### Comment · AnonReviewer3 · 2019-11-14
> > **Response**
> >
> > Thank you for your comments.
> >
> > 2. I did not mean to imply that you designed the network such that it could be published at ICLR. I'm sorry if it came out that way. I fully believe that this is the best network you found, and I very much appreciate that this is an important real-life task.
> >
> >
> > I've decided to retain my updated rating of weak reject.

---

> ### Author Response · Authors · 2019-11-14
> **Paper Revised 2**
>
> We once again thank Reviewer 3 for the extremely helpful comments.
>
> 1. In response to the comment on formalizing the problem, we completely rewrote the introduction to do a formal introduction of the problem and the theorem that proved DoS optimality (in a slightly different form to the original paper for easibility). Furthermore, we explicitly showed which terms we are approximating from the theorem.
>
> 2. Regarding the heuristic nature of the loss function, we added in the text a reference about how many products' CDF do not conform to well behaved parametric distributions.
>
> We would very much appreciate quick comments on the revision given the deadline for authors' responses are soon. We once again thank Reviewer #3 for the extremely helpful engagement on the paper.
>
> Best
> -Authors of Paper #924

---

### Official Review · AnonReviewer2 · 2019-10-27
**Official Blind Review #2**

**Rating:** 6

**Review:**

The paper introduces the duration-of-stay estimation problem in the warehouse storage application and describes a way to formulate the problem, prepare datasets, design loss functions and train models. The approach seems to significantly reduce the distance traveled in warehouses and therefore reduce labor costs.

This paper is an application paper where the problem studied seems to be limited to a small audience. I am not sure if it aligns well with ICLR. However, I do think the problem is interesting since it can directly lead to significant real-world improvements by improving the machine learning task. It is also nice to see the authors will publish the datasets to enable future research along this direction.

The paper is well written. It introduces the context of the problem very clearly, and outlines the important factors, and how machine learning can help solve the problem. I also like the detailed description of the problem formulation, datasets, and the loss function design.

I think the experiment section needs more work. For example, given that the input features are limited, can a GBDT do the job as well? Is neural network really needed here to learn a complicated representation? By comparing some simpler baselines, we can have a clearer idea into this question. Also, it is better if the paper can include some ablation studies to single out the important factors, for instance, which input features are most important, which kind of modeling enables the potential of which input features.

It is interesting the proposed approach seems to work well in real-world scenario. But it is not clear to me the comparison presented in Figure 3 really leads to increased efficiency of the warehouse. It is true more pallets are placed in nearby locations, but is the prediction of DoS accurate enough to prevent constantly retrieving pallets placed farther away? In other words, Figure 3 only describes the state after adopting the algorithm, but it does not say whether this configuration is better or not since one can come up with a totally random algorithm to place more pallets in nearby locations without leveraging the input features.


=============Update==============
After considering the rebuttal and the updated text, I have changed my rating to weak accept.

The authors have added more experiment comparisons and also some ablation studies to provide more insights of the problem. Also some clarification to Figure 3 is also added.

**Experience Assessment:**

I do not know much about this area.

**Review Assessment: Checking Correctness Of Derivations And Theory:**

I assessed the sensibility of the derivations and theory.

**Review Assessment: Checking Correctness Of Experiments:**

I assessed the sensibility of the experiments.

**Review Assessment: Thoroughness In Paper Reading:**

I read the paper at least twice and used my best judgement in assessing the paper.

---

> ### Author Response · Authors · 2019-11-06
> **Response to Review 2**
>
> Thank you Reviewer #2 for your comments.
>
> We have actually tested GBDT (was removed under the suggestion of a previous reviewer) and also linear regression models using frequency features and bag of words features from the textual data. the MSLE of GBDT was 1.1325 and that for LR is 0.9520 on the testing set, significantly underperforming the neural networks. We suspect the bad performance of GBDT is due to the highly categorical nature of product group information (>900 groups) and the fact that product group is very indicative of DoS.
>
> On real-life data, we argue that if it was a totally random algorithm that filled the warehouse as front as possible, the graph would be a concave 1-e^x type curve (this can be verified easily with a simulation) as the front positions would be very similar due to the randomness, and we would see a large dropoff at around the natural occupancy rate of the warehouse, which is about 70%. This is not the graph we see however, and fits better with the hypothesis that the pallets are arranged in increasing order of DoS, producing a 1/x type of curve.
>
> We also have additional evidence on the real data (since we are predicting a distribution, we can only compare distribution statistics): our average predicted duration is only about 15% away from the true one. We are happy to include this in the paper.

---

> ### Author Response · Authors · 2019-11-10
> **Paper Revised**
>
> Once again, thank you Reviewer #2 for the helpful comments.
>
> We have significantly updated the paper in response to your comments. This includes:
>
> 1. We added GBDT and LR model results into the experiments.
>
> 2. We added ablation studies to show the importance of certain features, and also the importance of both textual and nontextual features.
>
> 3. We added simulation plots for pallet placements for both an optimal DoS strategy and a greedy strategy that puts pallets in the nearest location possible without knowledge about its DoS. We show graphically that:
>
> 1. The DoS strategy is superior by putting more pallets into the front.
> 2. Our implemented DoS framework follows the DoS simulation graph relatively closely.
>
> This provides additional evidence that the proposed approach works well and generates real value to the warehouse.
>
> We sincerely hope you can update your review in light of these changes.
>
> Best
> -Authors of #924

---

> > ### Comment · AnonReviewer2 · 2019-11-14
> > **Updated review**
> >
> > I have updated the rating to "Weak accept" after considering the rebuttal and updated text.

---

> > > ### Author Response · Authors · 2019-11-14
> > > **Response**
> > >
> > > Thank you for your updated comments. We very much appreciate your engagement with us during this process, and are grateful for this.
> > >
> > > Best
> > > -Authors of Paper #924

---

### Decision · Program_Chairs · 2019-12-19

**Decision:**

Accept (Spotlight)

**Comment:**

Thanks to the authors for the submission and the active discussion. The paper applies deep learning to the duration-of-stay estimation problem in the warehouse storage application. The authors provide problem formulation and describe the pipeline of their solutions, including datasets preparation and loss functions design. The reviewers agree that this is a good application paper that showcases how deep learning can be useful for a real-world problem. The release of the dataset can also be a nice contribution. A major debate during the discussion is whether this paper is in scope of ICLR given that it is mostly a straightforward application existing techniques. After several rounds of discussion, reviewers think that this should fit under the category "applications in vision, ... , computational biology, and others." Overall, this paper can be a good example of applying deep learning to real-world problems.